# Geospatial Associations between Female Breast Cancer Mortality Rates and Environmental Socioeconomic Indicators for North Carolina

**DOI:** 10.3390/ijerph20146372

**Published:** 2023-07-15

**Authors:** Alanna Burwell, Sean Kimbro, Timothy Mulrooney

**Affiliations:** 1Department of Pharmaceutical Sciences, North Carolina Central University, Durham, NC 27707, USA; aburwel4@eagles.nccu.edu; 2Department of Microbiology, Biochemistry, and Immunology, Morehouse School of Medicine, Atlanta, GA 30310, USA; skimbro@msm.edu; 3Department of Environmental, Earth and Geospatial Sciences, North Carolina Central University, Durham, NC 27707, USA

**Keywords:** geospatial, breast cancer, socioeconomic, mortality, Geographic Information Systems

## Abstract

In North Carolina, over 6000 women will be diagnosed with breast cancer yearly, and over 1000 will die. It is well known that environmental conditions contribute greatly to health outcomes, and many of these factors include a geographic component. Using death data records from 2003–2019 extracted from North Carolina Vital Statistics Dataverse, a spatial database was developed to map and analyze female breast cancer mortality rates at the ZIP code scale in North Carolina. Thirty-nine hot spots and thirty cold spots of age-adjusted death rates were identified using the Getis–Ord analysis. Two-tailed *t*-tests were run between each cohort for environmental socioeconomic-related factors associated with breast cancer progression and mortality. The median age and household income of individuals who resided in ZIP codes with the highest breast cancer mortality were significantly lower than those who lived in ZIP codes with lower breast cancer mortality. The poverty rate, percentage of SNAP benefits, and the percentage of minorities were all significantly higher (*p* < 0.05, *p* < 0.001, and *p* < 0.001) in ZIP codes with high breast cancer rates. High-quality (ZIP code) granular cancer data were developed for which detailed analysis can be performed for future studies.

## 1. Introduction

Breast cancer (BCa) is a widespread disease that, unfortunately, leads to many deaths among women in the United States. Understanding the nature of this disease is crucial since there is a one in eight chance of developing it and a one in 39 chance of dying from it [1]. In North Carolina, the incidence rate for breast cancer is higher than the national average, as is the mortality rate [2]. Black women in North Carolina have an even higher mortality rate than other women in the state and the country. Despite significant advancements in breast cancer research, incidence rates continue to rise for Black women, who are more likely to be diagnosed with triple-negative breast cancer and experience a higher risk of mortality than White women [1,2,3,4,5,6,7]. 

The factors contributing to health disparities among Black women include racism, limited access to healthy food, lack of education, poverty, inadequate housing, lack of access to healthcare, environmental exposures, and criminal justice [8]. Systemic racism has been proven to create unjust power and resource distributions, significantly impacting an individual’s surroundings and ultimately affecting health outcomes [9]. Extensive research has shown that racism can adversely impact an individual’s health in numerous ways, such as restricting access to crucial opportunities, including education, employment, and housing, which can ultimately contribute to significant psychological stress [9,10]. These factors are connected geographically and are associated with an individual’s environment.

The physical spaces where individuals live, work, and engage in leisure activities are part of the built environment. Research has shown that restricted access to sidewalks, parks, green areas, and exposure to environmental pollutants in the community can negatively impact one’s overall health and wellbeing [9]. Studies have found that a lack of physical activity can lead to obesity, increasing the risk of developing other illnesses like BCa. Furthermore, psychological stress can result in unhealthy behaviors, such as reduced sleep and exercise, and increased smoking and alcohol consumption, which are factors that can initiate and worsen diseases [11,12,13,14].

Studies have revealed that psychological stress can contribute to the progression of BCa, especially in estrogen receptor negative (ER−) and triple-negative cases [15]. Further research has shown that stress hormones can damage DNA and affect cell cycle arrest, leading to the advancement of BCa [16]. Additionally, stress hormones like norepinephrine and epinephrine can interfere with the response of cancer cells to chemotherapy [17]. In BCa, stress hormones can increase the activity of cyclin-dependent kinase 1 (CDK1), advancing the cell cycle [18]. Activation of the adrenergic receptor (a receptor for norepinephrine and epinephrine) can cause gene transcription, cell survival, and proliferation, leading to nonresponsiveness or resistance to chemotherapy [19].

Stress can harm the body and lead to immune suppression, inflammation, epigenetic changes, and altered gene expression, all of which contribute to the development and progression of cancer, including BCa [20,21]. Studies have shown that women in low-SES neighborhoods are more likely to have ER-and triple-negative subtypes of BCa, which are more aggressive forms [22,23,24,25]. In fact, those in the lowest quartile of neighborhood SES have a 24% increased risk of ER-. Interestingly, women of low SES living in similar neighborhoods tend to have higher parity and earlier age at first birth, which are associated with a reduced risk of ER+ BCa, resulting in a better prognosis [26].

Public policy regulations acknowledge that the environment significantly impacts our health. Geographic Information Systems (GIS) is a valuable tool that combines data with visual representations to offer insights into the spatial distribution of health outcomes and their explanatory factors [27,28]. GIS has also been used to study sociodemographic factors and links between cancer-related diseases [27]. Using GIS to study social environmental-related stress factors can improve health outcomes, provide risk stratification, and close the BCa health disparities and mortality gap.

One way to gain insight into high breast cancer mortality rates is by generating cancer maps that highlight geographic variability. This method can help identify regions with elevated risks and draw attention to environmental factors contributing to stress and health disparities. To reduce these disparities for vulnerable populations, it is essential to determine which factors are associated with the environment and how to prevent exposure. As part of this effort for North Carolina, we aimed to identify, map, and characterize ZIP code areas with significantly high proportions of female breast cancer mortality rates using GIS. An essential aspect of this analysis involves extracting high-quality data (spatial, temporal, and attribute) directly from death records. These data can reveal patterns related to race, ethnicity, and age at a subcounty scale (ZIP code), and can be correlated with social and economic metrics collected at the same scale. By examining social and economic environmental factors associated with BCa mortality risk, this information will inform the development of public health initiatives to reduce breast cancer mortality rates.

## 2. Materials and Methods

### 2.1. Data Development and Mapping

Cancer mortality rates were extracted from the North Carolina Vital Statistics Dataverse (https://dataverse.unc.edu/dataverse/NCVITAL, accessed on 1 November 2022) for the years 2003 through 2019 [29]. Deaths for each year were provided as standalone tables via .data and .tab format that were converted to more conventional .csv and .xlsx files. In an average year, more than 100,000 death records that represent people who died in the state of North Carolina are stored using more than 40 attributes. Attributes include place of death, sex of decedent, date of death, state of death, county of death, birth county, birth state, race, ethnicity, marital status (if known), education level (if known), and cause of death using ICD-10 codes. Beginning in 2014, field names were changed, and fields related to tobacco use, pregnancy, and workplace injury were also added. 

Data were downloaded for each year, stored as a CSV (comma separated value) table and brought into GIS. Data for the years 2003 through 2013 were agglomerated into a single table, representing more than 1 million deaths during this time period. Data for the years 2014 through 2019 were “loaded” into the existing standalone table, using the scheme utilized in the 2003 through 2013 data. The field names for new data were mapped to the old field names. For example, the field name representing the birth county for the decedent is B_CNT for the older (2003–2013) data. However, in the new data schema provided from 2014 and afterwards, the birthplace for the decedent is BPLACE_CNT. As a result, all data from the BPLACE_CNT were imported into the B_CNT field for the master table. 

The result is a table encompassing 17 years of death records totaling more than 1.7 million records. From there, only female (SEX = 2) breast cancer (primary cause of death = C509) were queried from these records, resulting in 22,547 records representing all female breast cancer deaths between 2003 and 2019. While the actual locations or addresses of decedents were not stored in order to protect anonymity, the ZIP (Zone Improvement Plan) code typically used by the Postal Service is also stored as part of the schema provided by the Vital Statistics Dataverse to provide location. Using the GIS function Summarize, the number of deaths by ZIP code were counted and then joined to a spatial data layer representing North Carolina’s 763 ZIP codes. New fields within the ZIP code feature class were created, representing the number of deaths by ZIP code and the death rate (number of female breast cancer deaths per 10,000 female population) to be analyzed and mapped. 

### 2.2. Age-Adjusting Cancer Mortality

In some respects, raw numbers and crude death rates are excellent measures of mortality across a region, especially when allocating for or planning healthcare services; however, the major challenge with mapping raw cancer death rates (Figure 1) is they will correlate most closely with age (Buescher 2010). For the 725 ZIP codes that contain non-null values resulting from the Summary function, a simple correlation between median age and raw death rates results in an r^2^ = 0.147 and ρ < 0.001. As a result, data must be adjusted for age to control for differences among age distributions. Female breast cancer deaths were controlled for age using Buesher’s (1990, 2008, 2010) Statistical Primer published by the North Carolina State Center for Health Statistics [29].

In the age grading process, the raw number of female breast cancer deaths per ZIP code was queried out using the Select by Attribute and Summarize tool within the confines of the ArcGIS Pro 3.0 software for five separate age groups: 0 to 20 years old, 21–40, 41–60, 61–80, and 80+. While Buescher uses ten age cohorts in his work at the county level, given the larger ZIP code scale at which data are collected, larger age cohorts will decrease random variation in rates due to small numbers in some ZIP codes and age cohorts. After agglomerating the number of total female breast cancer deaths by age cohort and ZIP code and calculating their accompanying rates resulting in an age-specific death rate (ASDR), female population data for each ZIP code from the 2010 census data were imported. Using these population data, the percent of each age cohort was calculated against the total female population for each ZIP code, as shown in Table 1.

The age-adjusted death rate (AADR) was calculated using the Calculate Field functionality to be the ASDR weighted by the percent of population for each cohort, as shown in the formula below.
(1)$$0.264\times {ASDR}_{0–20}+0.264\times {ASDR}_{\;21–40}+0.272\times {ASDR}_{41–60}+0.162\times {ASDR}_{61–80}+0.038\times {ASDR}_{81+}$$

Using this formula, ZIP codes with high death counts due to the predominance of older populations, while contributing equally to raw death rates, will contribute much less to the age-adjusted death rates (AADR) because of the low proportion (3.8% for age 81+, for example) of older populations. Both these raw and AADRs can be mapped within the confines of GIS. 

Classifications of high and low AADRs using spatial techniques are based on the mean of all AADRs. Unreliably high or low values will skew the mean and, therefore, the definition of hot spots and cold spots. The same can also be said about aspatial comparisons, which will break the data into five equal quintiles. Because of this, all ZIP codes which contain a female population of 500 or less were removed. This decreased the number of working ZIP codes from 763 down to 715. Furthermore, it removed an inordinately high AADR value of 272 whose ZIP code only had a population of less than 30 women, where one or two deaths in this and other low-population ZIP codes will greatly skew AADRs. The removal of these 48 problematic ZIP codes decreased the mean of AADRs from 4.7 down to 4.3 and decreased the skewness of its histogram from 24.7 to 3.74. 

### 2.3. Aspatial Comparisons

As alluded to before, various socioeconomic–environmental variables were compared against high and low AADRs using a two-tailed *t*-test. Aspatially, high and low AADRs were computed based on quintiles. High AADR values will be defined as the highest 143 (upper fifth of 715 viable ZIP codes) AADRs values while low AADR values will be defined as the lowest 143 ZIP codes. 

### 2.4. Spatial Comparisons—Cluster Analysis

While data and maps have tremendous aesthetic value, geospatial statistics extend the capabilities of GIS and apply statistical principles to geospatial data to address questions related to statistical significance tied to location. Local Moran’s I is a local indicator of spatial autocorrelation (LISA) that calculates local variation based on adjacency patterns and an enumeration unit’s proximity to the mean of the dataset. An LISA returns an individual value for each enumeration unit, showing whether it is near other like values, in this case AADRs with a statistical significance. In the case of local Moran’s I, patterns that are similar and located near each other will return a high value for local Moran’s I. This is regardless of the fact that AADRs may be high or low. Expounding on local Moran’s I is another LISA that delineates the high–high and low–low relationships of clusters that are utilized in this research. The Getis–Ord Gi* uses a binary (1 or 0) neighborhood weight (defined by the user as adjacency or proximity *d* from the enumeration unit in question) to determine a weighted average of only those nearby values that satisfy the adjacency or neighborhood criteria. The end result is an inferential statistic that takes this Gi* calculation and returns a z-value, indicating a statistical significance of clustering in concert with a *p*-value assigned to an individual enumeration unit. The Getis–Ord Gi* was able to identify hot spots (high AADRs surrounded by other AADRs) and cold spots, representing low AADRs surrounded by other low AADRs with statistical significance [30].
$$\frac{G_{i}\left(d\right)=\sum_{j}w_{ij}\left(d\right)x_{j}}{\sum_{j}x_{j}}$$

### 2.5. Socioeconomic Data and the T-Test

To make associations between high/low vales values, as well as spatial clusters of age-graded death rates (AADR) and various socioeconomic–environmental measures (Table 2), socioeconomic–environmental data were extracted from the American Community Survey, National Land Cover Database (NLCD), and DataAxle, a data provider of businesses.

Each variable was run through an independent *t*-test between the two sets of samples. For spatial analysis, it is the ZIP codes of hot spots based on AADR versus the ZIP codes of cold spots based on AADR. For the aspatial analytical approach, it was the high AADR (defined to be upper quintile of AADRs across all eligible ZIP codes) versus the low AADR (defined to be lowest quintile of AADRs across all eligible ZIP codes). This *t*-test explores differences (in income as shown below, for example) between the 143 high AADRs and 143 low AADRs using this *t*-test. Using the sample size, mean, and standard deviation of the datasets in question, this *t*-test helped to determine the criteria in order to reject the null hypothesis and accept the alternate hypothesis using an example as shown below.

### 2.6. Null and Alternative Hypothesis for Aspatial Approach

$N_{0}$: Median household income from upper quintile of AADR = median household income from lowest quintile of AADR.

$N_{a}$: Median household income from upper quintile of AADR ≠ median household income from lowest quintile of AADR.

### 2.7. Null and Alternative Hypothesis for Spatial Approach

$N_{0}$: Median household income from hot spots of AADR = median household income from cold spots of AADR.

$N_{a}$: Median household income from hot spots of AADR ≠ median household income from cold spots of AADR.

## 3. Results

Death data records extracted from North Carolina Vital Statistics Dataverse were summarized for data related to female cancer mortalities in the state of North Carolina. A total of 22,547 records were extracted from an original database of more than 1.7 million records collected between the years 2003 and 2019. The age of death for these 22,547 records was 67.2 years. Figure 1A highlights the number of deaths each year from 2003 through 2019, with a high number of 1495 in the year 2017, and Figure 1B highlights the BCa mortality rate per 10,000 and the 3-year average from 2003–2019. In addition to listing these deaths by year, using the Summarize functionality within GIS (the equivalent of pivot tables in Excel), these deaths were cross-tabulated across various age, race, and birthplace variables based on fields captured in the North Carolina Vital Statistics Dataverse while running descriptive statistics of quantitative data. In Table 3, the race of each of the 22,547 decedents is highlighted by their average age, and Table 4 highlights the Hispanic ethnicity. It is apparent that African Americans have much lower average age of death than their White counterparts, with this age shown to be significantly different at the ρ < 0.001 level. 

### 3.1. Cluster Analysis

A ZIP code level map for crude female breast cancer death rates (per 10,000 women) is shown in Figure 2. As described previously, there are major challenges in mapping crude death rates given its relationship with age metrics such as median age. This is most pronounced in the western part of the state where the median age of residents is much higher. As a result, data representing AADR were calculated from ASDR and weighted based on the 2010 female population data for the state of North Carolina at the ZIP code scale. A map of these age-adjusted female breast cancer rates is shown in Figure 3. 

Distinct differences can be seen between figures, especially in the Research Triangle region of North Carolina. While crude death rates are relatively low in this region, AADRs are higher as a result of the age-adjusting process. This highlights that female breast cancer deaths impacts more populous age cohorts (21–60) in this region than older age cohorts (61+), thus bumping these numbers up from their crude death rates. 

Using the Getis–Ord Gi* statistic, hot spots of female AADR can be highlighted, showing high values surrounded by other high values, as well as cold spots which highlight low values surrounded by other low values. Since “high” and “low” are based on the mean AADRs of all values, unreliably high or low values were removed in the creation of the hot spot map shown in Figure 4 below. 

Thirty-nine hot spots of AADR were found from the Getis–Ord calculation and are most prominent in the eastern part of the state, with the strongest (>99% confidence) occurring mainly in the counties of Edgecombe, Northampton, Martin, Warren, Vance, and Robeson counties. While the 30 cold spots are spread throughout the state, the largest and strongest cold spots can be found in Onslow, Ashe, and Alleghany counties (Figure 4). While Onslow is one of the state’s youngest counties, both Alleghany and Ashe counties rank as the 9th and 13th (respectively) oldest counties in the state based on median age.

### 3.2. Aspatial and Spatial Comparisons

The lowest quantile of 143 breast cancer AADR by ZIP code was compared to that of the highest quantile of 143 AADR (Table 5). Two-tailed *t*-tests were run between each cohort for socioeconomic–environmental-related factors associated with breast cancer progression and mortality. The median age and household income of individuals who resided in ZIP codes with the highest breast cancer mortality were lower than those who lived in ZIP codes with lower breast cancer mortality. The poverty rate, percentage of snap benefits, and the percentage of minorities were all significantly higher (*p* < 0.05, *p* < 0.001, and *p* < 0.001) in ZIP codes with high breast cancer rates. The modified Retail Food Environment Index (mRFEI) was also significantly lower (*p* < 0.01) in the ZIP codes with higher breast cancer mortality. 

Two-tailed *t*-tests were also performed on the 30 cold and 39 high spots using the previously mentioned factors. The median household income was lower in the hot spots compared to the cold spots. The poverty rate was also significantly higher (*p* < 0.001) in the hot spots, as well as the percentage of minorities (*p* < 0.001) and the percentage of SNAP beneficiaries (*p* < 0.001). Furthermore, it was discovered that the mean quantity of vehicles in the hotspots was significantly greater (*p* < 0.01) than other areas (as shown in Table 6), indicating a possible link to elevated levels of air pollution.

## 4. Discussion

In this study, we sought to develop a spatial database to map and spatially analyze female breast cancer mortality rates at the ZIP code scale in North Carolina to delineate high and low mortality rates utilizing socioeconomic environmental factors. We used data from the 2003–2019 North Carolina Vital Statistics Dataverse to evaluate breast cancer age-adjusted death rates. This database consists of individual death records tagged with approximately 60 variables downloaded on a yearly basis. The end result is a database of more than 1.7 million records over this time period which can be queried for sex, race, ethnicity, and cause of death. Further complicating this database development was that the schema for individual attributes changed between the 2013 and 2014 editions of the data. New data attributes needed to be “mapped” to existing attribute. Data were able to be spatially analyzed within the confines of GIS using a ZIP code. As a result, age-adjusted female cancer mortality rates can be mapped across North Carolina’s 763 ZIP codes and analyzed using spatial statistics such as the Getis–Ord Gi* statistic. Using the Getis–Ord Gi* statistic, we verified cold and hot spots of breast cancer mortality rates. We showed hot spots of cancer mortality mainly located in the upper northeast region of North Carolina and cold spots in the northwestern and eastern areas of North Carolina. To make associations between high/low values and spatial clusters of AADR using various socioeconomic–environmental measures, we obtained data from the American Community Survey, National Land Cover Database (NLCD), and DataAxle. 

Consistent with other findings, we showed that Black women’s average mortality age was significantly lower than White women [31,32,33]. After performing two-tailed *t*-tests on socioeconomic environmental factors, we found the median household income to be significantly lower and the poverty rate significantly higher in ZIP codes marked as hot spots. Studies have shown that breast cancer mortality is higher in women of low SES, and women of lower SES were less likely to receive axillary surgery and adjuvant chemotherapy treatment [34]. Differences in receipt of breast cancer treatment could be due to limited access to nearby specialized providers. Research studies have also revealed that women who live in high-poverty ZIP codes are less likely to receive radiation than those who live in low-poverty areas [35,36].

This study is novel in the development of highly granular sub-county-scale cancer data that can be analyzed in concert with socioeconomic variable available at that same scale for the state of North Carolina. The master database developed and methodology highlighted in this project will go a long way in perpetuating the study of health disparities in North Carolina at spatial and temporal scales that can facilitate tangible change in our communities. An end result of this study revealed that a noteworthy proportion of minority communities live in ZIP codes with elevated breast cancer AADR. Research suggests that ethnic residential segregation may be responsible for placing minority populations in low-quality neighborhoods that lack necessary resources, leading to negative health consequences [37,38]. This could be due to racial or ethnic residential segregation, resulting in a difference in the geographic distribution of individuals by race or ethnicity [37,39,40,41].

Research has revealed that neighborhoods with high poverty rates have limited access to healthy food retailers, which can contribute to diets lacking essential nutrients and can increase the risk of chronic diseases [41,42]. In the United States, diet is a leading factor in chronic diseases like breast cancer [43,44]. Racial segregation in residential areas has been shown to decrease access to supermarkets [38] and increase access to fast-food restaurants and convenience stores that offer unhealthy food choices [45]. Although our study did not find a significant difference in distance to the nearest supermarket between low and high AADR ZIP codes, it did indicate a significantly lower modified retail food environment index (mRFEI) in ZIP codes with high breast cancer mortality. This suggests that there are more unhealthy food retailers in these areas compared to ZIP codes with lower AADR. Inadequate access to healthy food has been linked to negative impacts on BMI [42,46] and increased risk for breast cancer [47,48].

Our findings indicate that there is a significant difference in the average number of vehicles in hotspots versus cold spots. This disparity could suggest that increased vehicle pollution may be affecting the health of individuals in the affected areas. Studies have shown that air pollution from passenger vehicles can have negative impacts on respiratory health [49], as they emit various harmful pollutants such as carbon monoxide, sulfur oxides, oxides of nitrogen, and lead [50]. Both short-term and long-term exposure to air pollution can lead to a range of diseases, including stroke, pulmonary diseases, respiratory infections, and lung cancer. One study conducted by Chen et al. found that exposure to traffic-related air pollution significantly increased the risk of lung cancer [51,52,53]. In vivo, studies have also demonstrated that inhaling sulfur dioxide can lead to DNA lesions, which further increase the risk of developing cancer [54]. Another study, which examined occupational exposure to diesel exhaust, found that while there was no increased risk of overall breast cancer, diesel exhaust did elevate the risk of early-onset ER- tumors. This study emphasizes the need for further research into the relationship between air pollution exposure and the risk of ER- breast cancer [55].

In 2007, Edgecombe County in North Carolina was identified as having the highest incidence and mortality rates of breast cancer, leading to the formation of a task force to address the issue [56,57]. The task force implemented various initiatives, including raising community awareness, providing breast cancer resources, and enhancing existing community-based infrastructure, all of which contributed to the reduction of incidence and mortality rates [56]. Vines et al. suggested that the county’s economic development could also be beneficial in providing more jobs and better access to healthcare and health insurance [56]. Our analysis uncovered noteworthy differences in household income and poverty rates among BCa decedents in North Carolina. To bridge these gaps, policy changes, such as those mentioned earlier, are necessary to increase livable wage employment, improve education, and promote community development for healthier living environments. By implementing these policies, we can continue to reduce BCa mortality and other health disparities.

## 5. Conclusions

Many factors, both tangible and intangible, contribute to health disparities that can be observed over time and space. To better understand and address these disparities, Geographic Information Systems (GIS) can be an invaluable tool for mapping, analyzing, and visualizing health-related data. In this study, we analyzed a large database of deaths in North Carolina to map and spatially analyze breast cancer mortality rates at the ZIP code level. By identifying the highest and lowest breast cancer death rates, we were able to explore the relationships between socioeconomic–environmental variables at the ZIP code level. By using geostatistical methods, we were able to identify hot spots of breast cancer rates with statistical significance. Comparing socioeconomic–environmental variables between these hot spots and cold spots can help to identify disparities and target interventions to decrease breast cancer mortality rates. Our approach provides a targeted and effective way to address health disparities related to breast cancer. 

## Figures and Tables

**Figure 1 ijerph-20-06372-f001:**
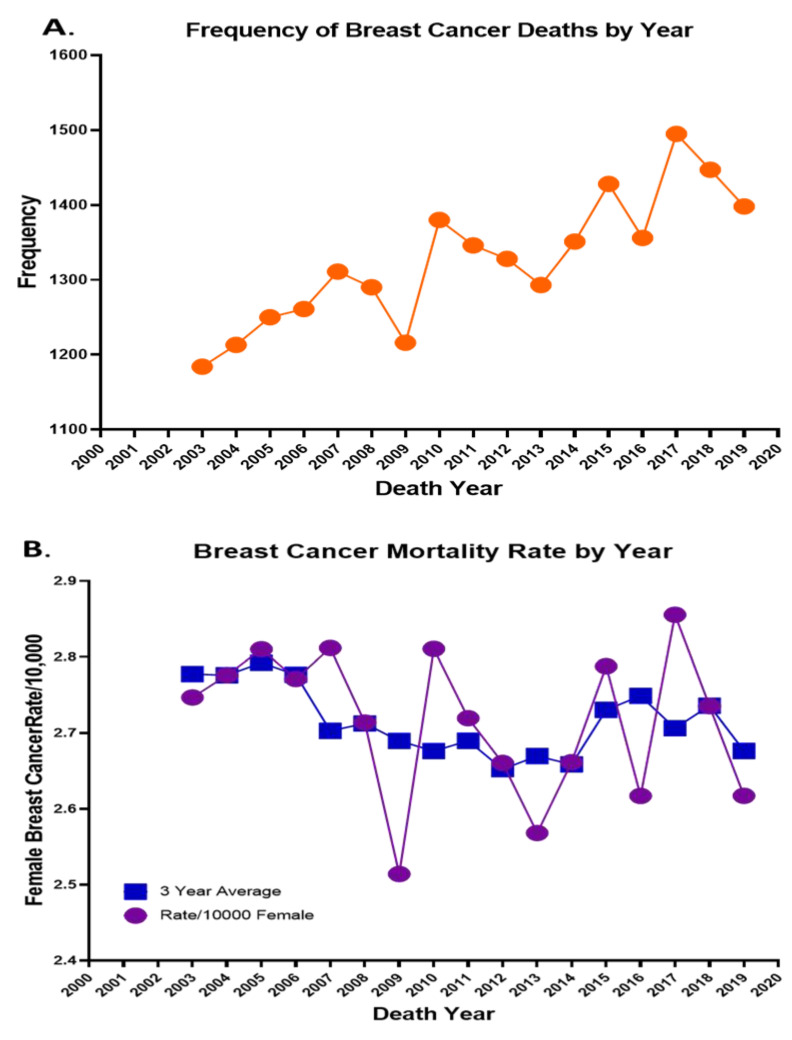
(**A**) Frequency of breast cancer deaths in North Carolina from 2003 to 2020. It also displays the 3-year average and yearly mortality rate for breast cancer (**B**).

**Figure 2 ijerph-20-06372-f002:**
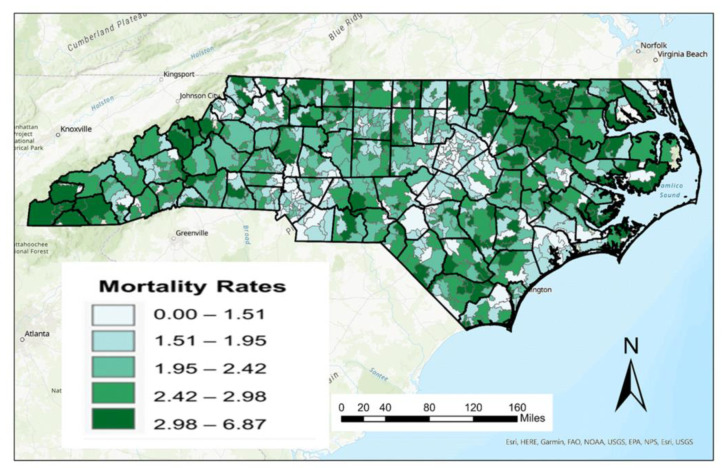
Raw female breast cancer mortality rate per 10,000 in North Carolina 2003–2019.

**Figure 3 ijerph-20-06372-f003:**
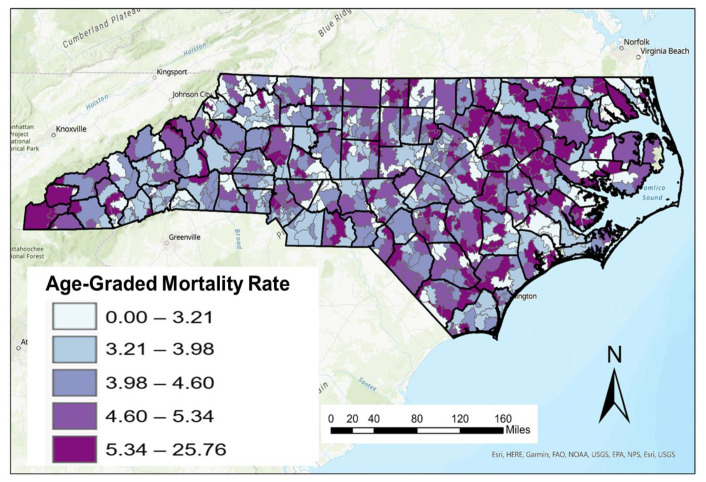
Age-adjusted female breast cancer mortality rate per 10,000 in North Carolina 2003–2019.

**Figure 4 ijerph-20-06372-f004:**
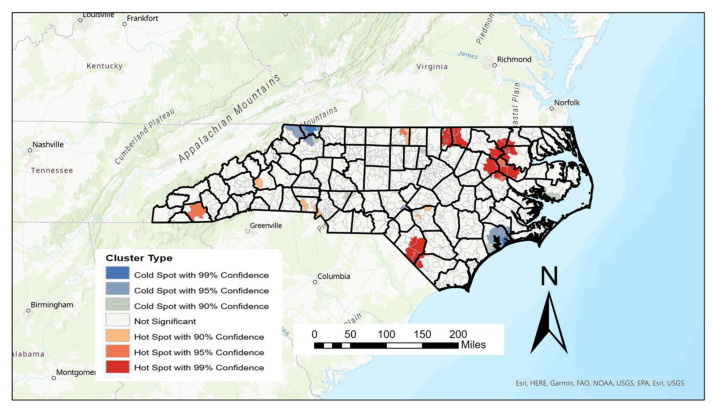
Cluster map of age-adjusted breast cancer mortality rates in North Carolina 2003–2019.

**Table 1 ijerph-20-06372-t001:** Percentage of age cohort.

Age Cohort	% of Population
0–20	26.4%
21–40	26.4%
41–60	27.2%
61–80	16.2%
81+	3.8%

**Table 2 ijerph-20-06372-t002:** Environmental–socioeconomic variables and descriptions.

Variable	Description
MED_AGE	Median age
MED_HH_INC	Median household income
PER_MINORITY	Percent minority population
PER_POV	Percent of households whose income is less than the poverty level
PER_POV2	Percent of households whose income is less than 200 percent of poverty level
PER_SNAP	Percent of residents who receive snap benefits
GREEN_SPACE	Percent of ZIP code composed of green space
mRFEI	Modified retail food environment—% of food outlets classified as healthy
DIST_SUPER	Distance from center of ZIP code to nearest supermarket
AVG_VEH	Average number of vehicles by household
DIST_HOSPITAL	Distance to nearest hospital (miles)

**Table 3 ijerph-20-06372-t003:** Race, number of deaths, and average age.

Race	Number	Average Age
White	16,345	68.85
African-American	5745	62.96
American Indian	196	64.07
Asian	256	58.21
Unknown	5	49.40

**Table 4 ijerph-20-06372-t004:** Number of deaths by ethnicity (Hispanic).

Ethnicity (Hispanic)	Number
Cuban	15
Mexican	58
Non-Hispanic	16,691
Other Hispanic	12
Puerto Rican	31
Central/South American	38
Unreported/Unknown	5702

**Table 5 ijerph-20-06372-t005:** Comparison of socioeconomic metrics of low and high aspatial breast cancer mortality.

	Low BCa Mortality	High BCa Mortality
# of enumeration units	143	143
MED_AGE	44.82 **	42.19 **
MED_HH_INC	46,939.24 *	43,693.53 *
PER_MINORITY	22.80278653 ***	37.39262501 ***
PER_POV	17.48315127 *	20.35663418 *
PER_POV2	40.60853 **	44.5699 **
PER_SNAP	14.33862113 ***	19.5670449 ***
GREEN_SPACE	4.15	3.85
mRFEI	22.80278653 **	15.54125 **
DIST_SUPER	4.15	3.85
AVG_VEH	2.35	2.39
DIST_HOSPITAL	13.81	11.95

* *p* < 0.05 ** *p* < 0.01 *** *p* < 0.001.

**Table 6 ijerph-20-06372-t006:** Comparison of socioeconomic metrics of low and high spatial breast cancer mortality.

	Low BCa Mortality	High BCa Mortality
# of enumeration units	30	39
MED_AGE	41.07	43.56
MED_HH_INC	46,049.70 *	40,022.18 *
PER_MINORITY	17.10 ***	49.27 ***
PER_POV	15.45 ***	21.49 ***
PER_POV2	41.80707	46.80082
PER_SNAP	12.37 ***	22.81 ***
GREEN_SPACE	4.18	4.37
mRFEI	16.84	16.86
DIST_SUPER	4.18	4.37
AVG_VEH	1.90 **	3.57 **
DIST_HOSPITAL	13.68	13.01

* *p* < 0.05 ** *p* < 0.01 *** *p* < 0.001.

## Data Availability

The datasets supporting the conclusions of this article are available in the Open Science Framework repository at https://doi.org/10.17605/OSF.IO/R8JWZ (accessed on 9 July 2023).

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
