# Peer review of "Geospatial Associations between Female Breast Cancer Mortality Rates and Environmental Socioeconomic Indicators for North Carolina"

_ijerph, 2023, doi:10.3390/ijerph20146372_

Round 1

Reviewer 1 Report

It is well known that the socio-economic status influences the survival and addition to therapy for cancer patients.

This study represents an interesting image of health disparity.

You stated (pg 9, 270): "axillary lymph node dissection may improve survival". Howver, it is previously shown that axillary surgery doesn't have any impact on survival for breast cancer patients. I would rather discuss the fact that not receiving radiotherapy is a factor that can alter survival for those patients with positive nodes and no other axillary treatment. 

Reviewer 3 Report

This is a very interesting article on decreasing breast cancer disparities in North Carolina. I think the use of the GIS and the inclusion of variables that measure the SES of the women who died from breast cancer across the state is an innovative approach toward portraying how geography, SES and cancer mortality are linked. The results of the study support existing literature on the topic of cancer  health disparities and therefore they can be used in the fight towards eliminating breast cancer disparities at a local level, one community at a time. However, I do have some comments and questions and hopefully my critique will strengthen the quality of the manuscript. 

A) Introduction section

* What is the epidemiological picture of breast cancer in North Carolina in terms of incidence variations based on race, screening rates by race and even information on stage of diagnosis ( localized, distant, or regional)? What are the trends and how does the epidemiological picture compare to that at the national level? It is important that the authors provide this information as a backdrop or the context where the study took place. 

* What are the  existing efforts at the state level to tackle breast cancer ( at all levels of prevention) and with emphasis on cancer disparities? How would the authors grade the existing efforts? What is the rationale of doing this study and how it relates to the existing efforts? 

* The authors state that stressful events, divorce, neighborhood crime are associated with higher breast cancer incidence. What is the mechanism that explains this association? What is the rationale behind this association? 

B) Methods

* Line 115. The authors indicate Table X. Where is Table X? 

* Line 191. The highest age of death from breast cancer was recorded to be 108. Is this a typo? It sounds a little bit incredible that at this age someone will die from breast cancer. Just curious..

* Figure 1 shows that overall there is a positive trend from years 2003 to 2020 on the frequency of breast cancer deaths in North Carolina. However, if you look at the figures from the recent report from the American Cancer Society ( Facts and Figures) the overall trend in breast cancer mortality during the last 30 years by race has remained relatively stable or decreasing ( with the exception perhaps of the American Indian community). How comes the mortality rate or the frequency of breast cancer deaths is increasing in North Carolina? If it is true, this is very alarming and the authors need to provide some explanation. 

* Table 2: I think the authors forgot to include the Hispanic population in their calculations. It is estimated that 11% of the population in North Carolina are Hispanics. It is hard for me to believe that Hispanics are included under the “unknown” category. 

* Why focus on mortality rates and not on incidence rates? 

C. Discussion

* In line 250 it was mentioned that the highest breast cancer mortality was found in zip codes with the highest number of vehicles. Why is this a significant finding and how it relates to the discussion of lower SES and higher breast cancer mortality? 

* I think the authors provide some good justifications for the results. However, I am wondering if other factors such as rural vs. urban areas, cultural related reasons, low vs high screening rates can also play a role in explaining the differences in the breast cancer mortality rates and how those factors could relate to the SES variables measured for this study.

* How will this information be useful in developing policies at the local level to close the gap among groups of different SES levels in terms of breast cancer mortality and incidence? How can the results of the study be translated into programs or strengthen the existing efforts of the state of North Carolina to tackle breast cancer? I think more information on the use of the results should be included in the discussion. 

* What are the limitations of the study? 

Round 2

Reviewer 2 Report

The authors made efforts to respond to all questions addressed and the quality and clarity of the paper are now considerably improved -and worths publication.